# Water Balance Calculation Based on Hydrodynamics in Reservoir Operation

**Sipeng Zhu** [1], **Shuo Ouyang** [2], **Jianzhong Zhou** [1,*], **Hongya Qiu** [1,3], **Hui Qin** [1] , **Jingwei Huang** [1] **and Xinqiang Niu** [4]

1. School of Civil and Hydraulic Engineering, Huazhong University of Science & Technology, Wuhan 430074, China; d201880951@hust.edu.cn (S.Z.); hongya.qiu@foxmail.com (H.Q.); hqin@hust.edu.cn (H.Q.); d202180555@hust.edu.cn (J.H.)
2. Bureau of Hydrology, Changjiang Water Resources Commission, Wuhan 430010, China; niuxinqiang@cjwsjy.com.cn
3. China Three Gorges Corporation, Yichang 443133, China
4. Changjiang Institute of Survey, Planning, Design and Research Corporation, Wuhan 430010, China; papercommunicate@163.com
* Correspondence: jz.zhou@hust.edu.cn

**Abstract:** Reservoir operation plays an important role in reservoir management. In reservoir operation, water balance calculation is a very important step. At present, one of the main challenges is that reservoir inflow cannot be calculated accurately due to jacking of the reservoir, which is produced by a downstream reservoir after the original course of the river has changed. Another reason that reservoir capacity cannot be calculated accurately is due to the influence of dynamic storage capacity. In order to overcome these problems, this report shows that the land zone in front of the dam can be used to calculate reservoir capacity, since it can serve as the boundary for a hydrodynamic model, which can then be used to calculate reservoir inflow to improve accuracy. The Three Gorges Dam was selected as a case study. The results show that compared with the measured data, the RMSE (root mean square error) of the proposed model was 8.5%, whereas the RMSE of the traditional model was 25.9%. The contributions and novelty of this paper are: (a) the proposed model combines a hydrodynamic model with a water balance calculation model to make the calculation of inflow more accurate; (b) the land zone in front of the dam can be used to make the calculation of reservoir capacity more accurate; (c) the proposed method provides a new way to calculate water balance, which can be used for short-term reservoir operation. The application results indicate that this study can provide technical support for the fine operation of reservoirs.

**Keywords:** dynamic capacity; land zone in front of the dam; hydrodynamic model; water balance model; reservoir operation



## 1. Introduction

With the completion of cascade hydropower stations in China and other countries all over the world, attention has gradually shifted to focus on improving the operation effectiveness and efficiency of existing reservoir systems to maximize the beneficial uses of these projects [1,2]. In the past 20 years, a series of hydraulic projects have been built and put into power generation, which has promoted the development and utilization of water resources [3,4]. At the same time, with the goal of energy conservation and emission reduction, the demand for clean energies has expanded greatly. Hydropower is one of the most important clean energies [5,6] that will become more important in the future [7]. With the continuous improvement of the status of hydropower in the power grid, demand for the fine operation of reservoirs has increased. In reservoir operation models, reservoir inflow is a very important input factor, affecting the accuracy of the reservoir dispatching schedule.

Inflow calculation methods, also known as flow propagation, can be divided into two categories, hydrological methods and hydrodynamic methods. The lag time method and Muskingum method are the most commonly used hydrological methods for flow propagation [8]. The lag time method only considers the time delay, which delays the outflow from the upper reservoir for a period of time without changing its value. However, flow propagation in a river can be divided into two parts: one is flow transposition and the other is flow attenuation [9]. Due to downstream reservoir jacking and flow attenuation, the lag time is not a constant, and this will lead to a large deviation in the calculation results. To solve this problem, Zhang and Liu [10] adopted data mining methods to find the relationship between the outflow of the upstream reservoir and the inflow of the downstream reservoir. The training of the model needs a large amount of data, which is hard to obtain, the interpretability of the model is low, and the effect in actual application is difficult to guarantee. The Muskingum method was used to overcome the disadvantage of the lag time method. The Muskingum model was proposed by Carthy [11] in 1938, and has been widely used in hydrology research. This model is derived from the Saint-Venant equations, which is suitable for natural channels [12]. Ji [9] used the Muskingum method to solve the problem of large flow calculation errors caused by lag time method. Many studies have shown that this method can solve the error problem caused by the lag time method [8,10]. However, when the river is jacked by a downstream reservoir, the linear tank storage assumption of the Muskingum method will fail, resulting in a calculation error [13]. The Muskingum method is a simplification of the one-dimensional hydrodynamic control equation. In order to solve the problem when the river is supported by a downstream reservoir, it is necessary to solve the complete one-dimensional hydrodynamic control equation for a more accurate calculation of reservoir inflow [12]. The fine operation of a reservoir requires an accurate calculation of inflow. Therefore, it is necessary to combine the one-dimensional hydrodynamic model with the reservoir operation model to achieve fine operation of the reservoir. The control equation of the one-dimensional hydrodynamic model is also called the Saint-Venant equations [14]. The one-dimensional Saint-Venant equations are hyperbolic partial differential equations [15], but they have no analytical solution, so many numerical methods have been proposed to solve the equations in the past few decades. Abbott presented a six-point implicit method to solve the Saint-Venant equations, which is used by DHI MIKE [16]. DHI MIKE is widely used to calculate flow propagation, but commercial software is not open source, and the computational efficiency of MIKE software is very low, which makes unsuitable for coupling with the reservoir operation model. The Riemann solver is another way to solve the Saint-Venant equations [17,18], but these methods are affected by CFL (Courant–Friedrichs–Lewy) conditions [19–21] so that the Riemann solver also has low computational efficiency. This disadvantage seriously restricts coupling the hydrodynamic model with the reservoir model. To overcome the low computational efficiency of the traditional hydrodynamic model, a hydrodynamic model using the ELM (Eulerian–Lagrangian approach) method was developed by Casulli [19]. The advantage of this model is that it is robust and efficient. Relevant tests show that a hydrodynamic model using the ELM method is 3~6 times faster than traditional hydrodynamic models [20,21]. This advantage makes it possible to couple the hydrodynamic model and the reservoir operation model. Hu et al. [20] proposed a hydrodynamic model using ELM to simulate a river network, which not only has these advantages, but also further improves the computational efficiency and robustness of the algorithm.

Furthermore, fine operation of a reservoir also requires an accurate calculation of its capacity. However, traditional reservoir operation models calculate capacity based on a level-capacity curve, which assumes that the water level is straight for the whole reservoir area. This method will cause large errors in the reservoir capacity due to the dynamic capacity. Because the water surface line of the reservoir is determined by geophysical factors [22,23] and the season, the water surface line for the reservoir in mountainous country cannot be ignored during the flood season. In view of this situation, the concept of the land zone in front of the dam was proposed [24]. Using the gradient of the water surface

line [25,26], we divided the reservoir area into two parts, one for the land zone in front of the dam, and the other for a natural channel. The strictly straight land zone obeys the assumption of a static capacity curve, so the reservoir capacity can be calculated accurately.

In this paper we combine a hydrodynamic model with a reservoir operation model. The specific method is as follows: the land zone in front of the dam is used to establish a water balance calculation model, realizing accurate calculation of reservoir capacity through the straight section in front of the dam, and avoiding the influence of dynamic storage capacity; at the same time, a hydrodynamic model is used to provide inflow for the reservoir operation model for realizing accurate calculation of reservoir inflow.

The paper is organized as follows: An overview of the models used in this paper is provided in Section 2. In Section 3, the Three Gorges Reservoir area is used to verify the model proposed in this paper, and the data used in the research is introduced. Section 4 uses the Three Gorges Reservoir as the research object for discussion and analysis. Section 5 summarizes the research of the paper.

## 2. Methods

### 2.1. The Hydrodynamic Model

The governing equation for a one-dimensional hydrodynamic model is simplified from a two-dimensional shallow water equation. The governing equations include a continuity equation and a momentum equation and are given by:

$$B\frac{\partial \eta}{\partial t} + \frac{\partial Au}{\partial x} = q \tag{1}$$

$$\frac{\partial u}{\partial t} + u\frac{\partial u}{\partial x} = -g\frac{\partial \eta}{\partial x} - g\frac{n_m^2 u|u|}{R^{4/3}} \tag{2}$$

where $B$ is the width of the section water surface (m); $\eta$ is the central water level of the section (m); $u$ is the cross-sectional averaged velocity (m/s); $A$ is the cross-sectional area of water discharge (m$^2$/s); $g$ is gravitational acceleration (m$^2$/s); $n_m$ is the comprehensive roughness value for the river channel (m$^{-1/3}$/s); $R$ is the hydraulic radius of the section (m); $t$ is the calculation time step (s); $x$ is the spatial step(m).

Here, we use a set of 1D cells to obtain a graphical description of the river. An unstructured staggered grid with a variable arrangement is used to discretize the equation (see Figure 1). The cross-section topology data is arranged at the center of the control volume. The control volume has at least two interfaces (see dotted line in Figure 1). The velocity $u$ is defined at the interface centers, and the water level $\eta$ is defined at the cell centers. The biggest advantage is that, compared with a collocated grid, a staggered grid can maximize the use of measured terrain data without reconstructing the interface for interpolation. The control volumes are sequentially donated by $i, i + 1,..., N$.

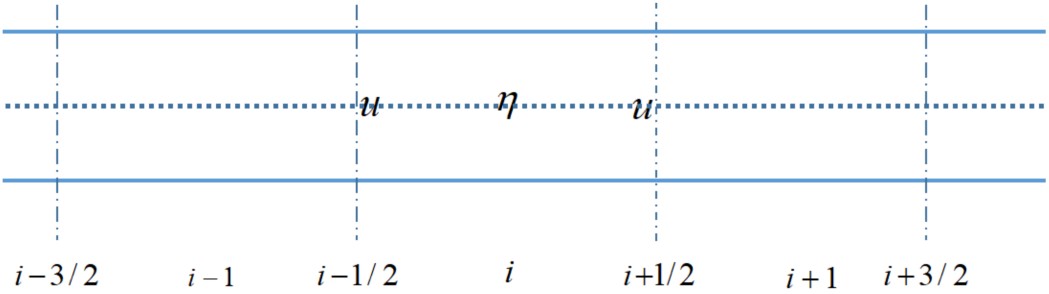

**Figure 1.** The control volume and grid used in the hydrodynamic model.

To further describe the unstructured staggered grid, the following variables are introduced: (1) $I34(i)$ is the number of interfaces of control volume $i$; (2) $J(i,l)$ is the interface of the control volume; (3) $IC(i,l)$ are the cells that have common interface; (4) $s_{i,l}$ is a sign

function that represents the positive direction of interface velocity and is calculated by the following equation:

$$s_{I,l} = \frac{I[J(I,l),2] - 2I + I[J(I,l),1]}{I[J(I,l),2] - I[J(I,l),1]}$$

### 2.1.1. Discretizing the Momentum Equation

The momentum equation is discretized using the operator-splitting [14,27,28] technique. The advection term is solved by ELM. To eliminate the stability restriction caused by gravity waves, the semi-implicit method is introduced to discretize the water-level gradient term. After discretizing, the water-level gradient is as follows [20]:

$$\left(1 + g\Delta t \frac{n_m^2 |u_{bt,i+1/2}^n|}{R_{i+1/2}^{4/3}}\right) u_{i+1/2}^{n+1} = u_{bt,i+1/2}^n - (1-\theta)g\Delta t \frac{\eta_{I(i+1/2,2)}^n - \eta_{I(i+1/2,1)}^n}{\delta_i}$$
$$-\theta g\Delta t \frac{\eta_{I(i+1/2,2)}^{n+1} - \eta_{I(i+1/2,1)}^{n+1}}{\delta_i} \tag{3}$$

where the $u_{bt,i+1/2}^n$ is obtained by the ELM method; $\delta_i$ is the length of the control volume; $\theta$ is the implicit coefficient, which is between 0 and 1 [19]; $\Delta t$ is the calculation time step. Equation (3) can be transformed into the following form:

$$u_{i+1/2}^{n+1} = \frac{G_{i+1/2}^n}{F_{i+1/2}^n} - \theta\Delta t g \frac{1}{F_{i+1/2}^n} \frac{\eta_{i+1}^{n+1} - \eta_i^{n+1}}{\Delta x_{i+1/2}} \tag{4}$$

where $F_{i+1/2}^n = 1 + g\Delta t n_m^2 |u_{bt,i+1/2}^n| / R_{i+1/2}^{n4/3}$ and $G_{i+1/2}^n = u_{bt,i+1/2}^n - (1-\theta)g$ $\Delta t \frac{\eta_{I(i+1/2,2)}^n - \eta_{I(i+1/2,1)}^n}{\delta_i}$.

### 2.1.2. Discretizing the Continuity Equation

In the continuity Equation (1), the term $\partial\eta/\partial t$ can be descretized as $B^n(\eta^{n+1} - \eta^n)$. For any control volume, the continuity equation can be rewritten as:

$$B_i^n \Delta x_i \left(\eta_i^{n+1} - \eta_i^n\right) = -\theta\Delta t \sum_{l=1}^{I34(i)} s_{i,l} A_{J(i,l)}^n u_{J(i,l)}^{n+1}$$
$$-(1-\theta)\Delta t \sum_{l=1}^{I34(i)} s_{i,l} A_{J(i,l)}^n u_{J(i,l)}^n + \Delta t \Delta x_i q_i^n \tag{5}$$

After discretizing the continuity and momentum equations, we need to solve the velocity-pressure coupling problem. For one river (see Figure 1), velocity-pressure is performed by substituting Equation (4) with Equation (5), then we can get Equation (6).

$$C_i^a \eta_{i-1}^{n+1} + C_i^b \eta_i^{n+1} + C_i^c \eta_{i+1}^{n+1} = r_i^n \tag{6}$$

where $C_i^a = -\frac{g\theta^2\Delta t^2 A_{i-1/2}^n}{F_{i-1/2}^n \Delta x_{i-1/2}}$, $C_i^b = B_i^n \Delta x_i + \frac{g\theta^2\Delta t^2 A_{i-1/2}^n}{F_{i-1/2}^n \Delta x_{i-1/2}} + \frac{g\theta^2\Delta t^2 A_{i+1/2}^n}{F_{i+1/2}^n \Delta x_{i+1/2}}$, $C_i^c = -\frac{g\theta^2\Delta t^2 A_{i+1/2}^n}{F_{i+1/2}^n \Delta x_{i+1/2}}$ and

$$r_i^n = -\theta\Delta t \left[A_{i+1/2}^n G_{i+1/2}^n / F_{i+1/2}^n - A_{i-1/2}^n G_{i-1/2}^n / F_{i-1/2}^n\right]$$
$$-(1-\theta)\Delta t \left[A_{i+1/2}^n u_{i+1/2}^n - A_{i-1/2}^n u_{i-1/2}^n\right] + B_i^n \Delta x_i \eta_i^n + \Delta t \Delta x_i q_i^n$$

For a river after velocity-pressure coupling, we can get a tridiagonal linear system. This system has a symmetrical and positive definite coefficient matrix and can be solved directly without iterations. The system can be solved by Thomas algorithm. See the literature [14] for the detailed process.

2.1.3. The Difference between the Muskingum Model and Hydrodynamic Model

In most of the hydrology literature, the control equation for the Muskingum method is obtained in this way [12]:

$$V_2 + \frac{\Delta t}{2} Q_2 = \frac{\Delta t}{2}(I_2 + I_1) - \frac{\Delta t}{2} Q_1 + V_1 \tag{7}$$

$$V = K[xI + (1-x)Q] \tag{8}$$

Coupling Equations (7) and (8) yields the following formula:

$$Q_2 = C_1 I_1 + C_2 I_2 + C_3 Q_1 \tag{9}$$

Equation (9) is the control equation for the Muskingum method [29].

However, there is another view of how to derive this equation. Cunge [17] proposed beginning with the following equation:

$$\frac{\partial Q}{\partial t} + c\frac{\partial Q}{\partial x} = 0 \tag{10}$$

where $c$ is the wave speed that is calculated from $c = \frac{1}{n_m} R^{2/3} S_0^{1/2} + A\frac{1}{n_m} S_0^{1/2} \frac{d(R^{2/3})}{dA}$, $Q$ is the discharge (m$^3$/s), $x$ is the longitudinal coordinate. Cunge derived the following classical finite difference weighted approximation for the partial derivatives of a four point scheme.

$$\frac{\partial Q}{\partial t} \approx \frac{\varepsilon(Q_j^{i+1} - Q_j^i) + (1-\mu)(Q_{j+1}^{i+1} - Q_{j+1}^i)}{\Delta t} \tag{11}$$

$$\frac{\partial Q}{\partial x} \approx \frac{\theta(Q_{j+1}^{i+1} - Q_j^{i+1}) + (1-\mu)(Q_{j+1}^i - Q_j^i)}{\Delta x} \tag{12}$$

where the $\mu$ is the space weighting factor, which is between 0 and 1, and $\varepsilon$ is the time weighting factor, which is between 0 and 1. Equations (11) and (12) can lead to the following first order approximation of the kinematic wave equation:

$$\frac{\varepsilon\left(Q_j^{i+1} - Q_j^i\right) + (1-\varepsilon)\left(Q_{j+1}^{i+1} - Q_{j+1}^i\right)}{\Delta t} + c\frac{\theta\left(Q_{j+1}^{i+1} - Q_j^{i+1}\right) + (1-\mu)\left(Q_{j+1}^i - Q_j^i\right)}{\Delta x} = 0 \tag{13}$$

Assuming that $\theta$ is equal to 0.5 [17], Equation (13) can be rewritten as

$$\frac{\varepsilon\left(Q_j^{i+1} - Q_j^i\right) + (1-\varepsilon)\left(Q_{j+1}^{i+1} - Q_{j+1}^i\right)}{\Delta t} + \frac{c}{2}\frac{\left(Q_{j+1}^{i+1} - Q_j^{i+1}\right) + \left(Q_{j+1}^i - Q_j^i\right)}{\Delta x} = 0 \tag{14}$$

After combining and simplifying formula (14), Equation (9) can be obtained.

From the derivation process, we can see that the Muskingum method is not strictly satisfied with the momentum formula [12]. When the river is a natural channel, the flow propagation is not impacted by the water-level gradient term, which is ignored by the Muskingum method. However, if there is a reservoir downstream of the river, the flow propagation will be seriously impacted by the water-level gradient method. In this situation, the Muskingum method will produce large error.

*2.2. The Water Balance Calculation*

2.2.1. The Land Zone in Front of the Dam Section

The water surface of river type reservoirs in mountainous areas will form a certain gradient, so the actual water surface is not horizontal, as shown in Figure 2 [30], which leads to the existence of dynamic storage capacity in the reservoir. The water surface in the reservoir area is divided into the land zone in front of dam and a warped section. The accuracy of the calculation results can be guaranteed when using a static storage capacity

curve to calculate the storage capacity of the land zone in front of dam section, but the storage capacity will be smaller when calculating the warped section.

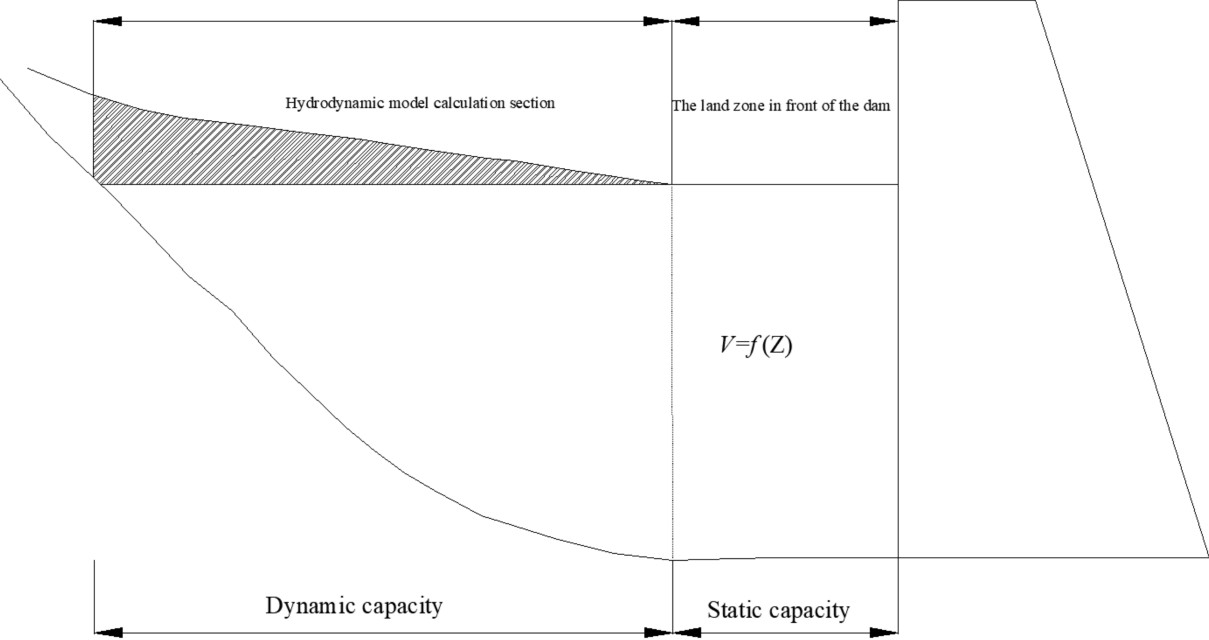

**Figure 2.** Schematic diagram of the straight section in front of a dam.

The length of the land zone in front of dam is not a constant. There are many factors affect the length of the land zone in front of the dam, such as the water level in front of the dam and inflow discharge.

2.2.2. Combining the Water Balance Calculation and Hydrodynamic Model through the Straight Zone in Front of the Dam

Taking the land zone in front of dam section as the medium, the hydrodynamic model from the upstream prediction station to the end of the land zone in front of the dam is established. The new water balance calculation model proposed in this paper uses the land zone in front of dam to establish a local control body for the water balance calculation. The calculation steps are given below as Figure 3, and all steps are completely automatic:

Step1: determining the range of the land zone in front of the dam by using the hydrological data and the calculation time (T).

Step2: establishing the hydrodynamic model between the upstream hydrological station and the end of the straight zone in front of dam using the measured domain data to establish the hydrodynamic model.

Step3: using the new model to deal with the water balance calculation.

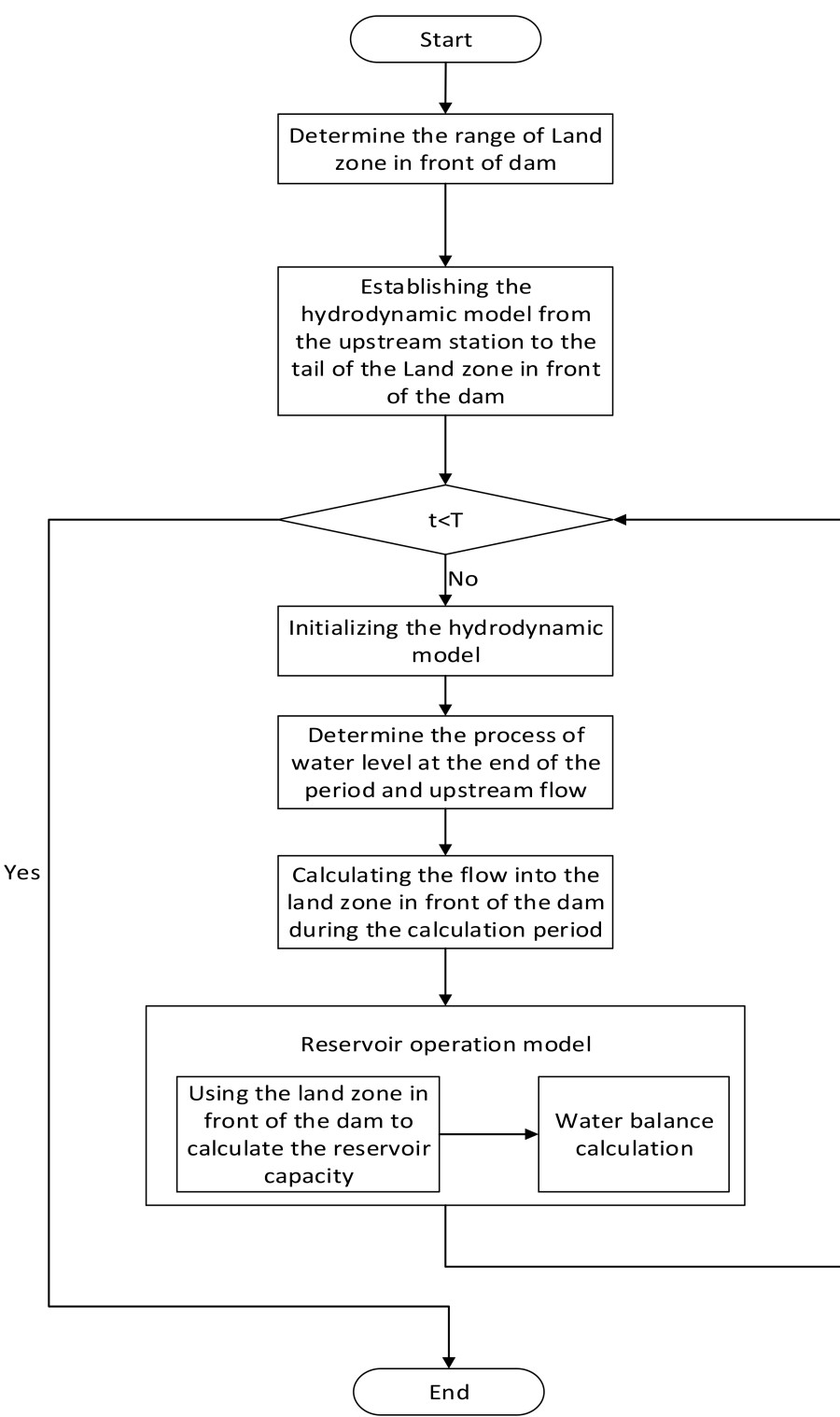

**Figure 3.** The calculation steps for the new water balance model.

## 3. Study Area and Data

### 3.1. Study Area

The Three Gorges Reservoir area, which is located downstream of the Jinsha River was selected as the study area, stretching from Zhutuo to the Three Gorges Dam. The length of the study area is about 760 km. The width of the Jinsha River is between 700 and 1700 m. The Three Gorges Reservoir area is located at the alternation of the second and third levels of topography in China. The geographical conditions are very complex in the

Three Gorges Reservoir area, which is surrounded by the Qiling and Daba Mountains in the north and Wu Mountain in the south, and the area comprises 74% mountains, 21.7% hills and 4.3% plains [31]. It is a typical river-type reservoir, and therefore, the Three Gorges Reservoir has great dynamic capacity. The Three Gorges Reservoir area begins at the hydrological station in Zhutuo, as Figure 4 shows, and along the river, there are many hydrological stations, including Cuntan, Qingxichang, Wanxian and Fenghuangshan. The Fenghuangshan hydrological station is close to the dam, so we can use the data from this station on behalf of the water level in front of the dam. There are many small rivers in the reservoir area, such as Quxi River and Modao River, among others. The reservoir area has a subtropical monsoon climate, and the large amount of precipitation in the flood season will lead to interval runoff that cannot be ignored for the whole interval. Relevant studies show that the average contribution of interval inflow to the flood peak in the reservoir area is 20%, and the maximum is close to 50% [32]. The total storage capacity of the Three Gorges Reservoir is 39.3 billion m$^3$, and the flood control storage capacity is 22.15 billion m$^3$. The Three Gorges hydropower station is the largest hydropower station in the world, with a total installed capacity of 22.5 million kW that is produced by thirty-two 700,000 kW generating units and two 50,000 kW auxiliary power units. The normal water level of the Three Gorges Reservoir is 175 m, and the flood limited water level is 145 m.

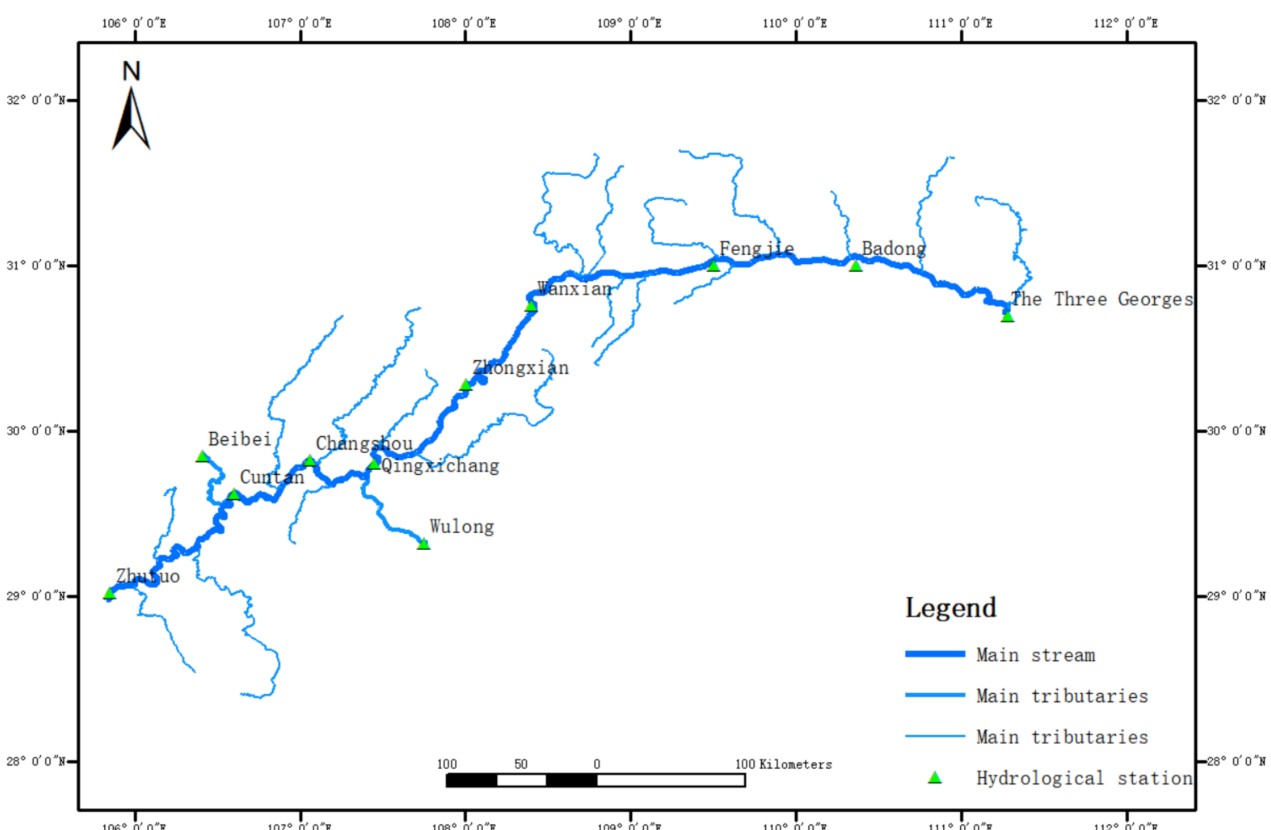

**Figure 4.** Schematic diagram of the study area.

*3.2. Data*

To compare the Muskingum method and hydrodynamic model, we used the measured hydrological data from 2005. This is because construction of the Three Gorges Dam was finished in 2006. After the dam was finished, the hydrological station in front of the dam could not measure flow in the river. Similarly, the year the dam was nearly finished also had an impact on the river. Thus, we can use the data from 2005 to compare the differences between the two methods.

To validate the new water balance model proposed in this paper, we used data from 2016 to calibrate the parameters and data from 2017 to test the accuracy of the model parameters. The data used in this article are measured hydrological data that have been reviewed for reliability, consistency and representativeness prior to use. To verify the accuracy of the model with actual operation data, the hourly scale data from 5 July 2019 to 10 July 2019 were combined with the straight section in front of the dam to verify the accuracy of the regulation and storage model.

## 4. Case Study

### 4.1. The Comparison between the Hydrodynamic Model and the Muskingum Method

This paper uses Miaohe hydrology station, which is in front of the dam, to compare the two methods. As we can see from Figure 5, there is a large error between the Muskingum method and the measured values. It can be seen from the figure that in the case of multi-peak floods, the Muskingum method and hydrodynamic method are close to the measured values for the first peak, but the flow values from the Muskingum method decreased significantly later. This is because the Muskingum method is a free flow hypothesis, but after the dam was completed, the free flow hypothesis was no longer valid due to the jacking effect of the dam. The average relative error for the Muskingum method was 10.24%, however the average relative error for the hydrodynamic model was 2.79%. During flood season, the average relative error for the Muskingum method was 12.96% compared to 3.57% for the hydrodynamic model. The dam was not completed in 2005, but the Muskingum method still had a larger error than the hydrodynamic method. After the completion of the dam, jacking of the downstream water level was more obvious, so the Muskingum method will no longer be applicable.

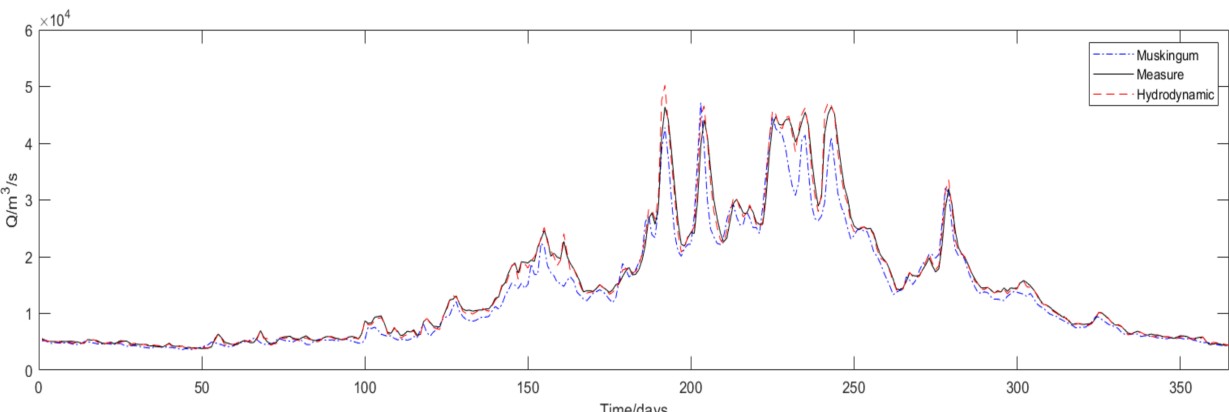

**Figure 5.** Comparison of the results from the Muskingum method and the hydrodynamic model with measured data from 2005.

### 4.2. Validation of the Improved Water Balance Calculation Model

After the completion of the dam, the parameters of the hydrodynamic model needed to be calibrated again. The data from 2016 was chosen for calibration and 2017 data was selected to verify the parameter results. The calibration results are shown in Figure 6.

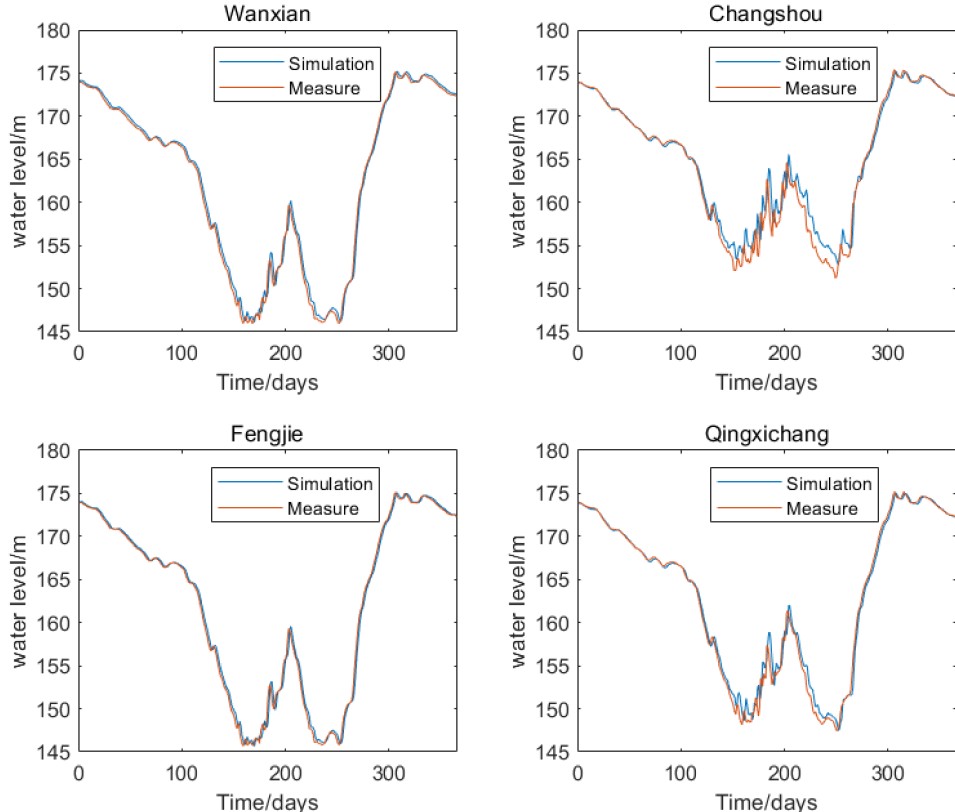

**Figure 6.** Comparison of the results from the simulation with measured data from 2016.

As can be seen in Figure 6, the closer the hydrological station is to the Three Gorges Dam, the more accurate the simulation results. It can be seen from Table 1 that the relative error for Changshou hydrological station is the largest, reaching 0.5%. It can be seen in Figure 4 that Changshou hydrological station is located at the tail of the Three Gorges Reservoir area. Changshou hydrological station is affected by the dynamic capacity during the flood season, and its water level fluctuates greatly. It can be seen in Figure 6 that the error is more obvious during the flood season, whereas the simulated and measured values are in good agreement during the non-flood season. This is because the manning coefficient differs with the different water levels at Changshou hydrological station, which affects the simulation results. It can be seen from Table 1 that the minimum relative error for the four stations is 0. This is because the Three Gorges Reservoir continues to operate at a water level of 175 m during the dry season and the flow velocity in the entire reservoir area is slow, so the relative error is very small. The calibration results for the Manning coefficient are shown in Table 2.

**Table 1.** Statistical comparison results for simulated and measured data at different hydrological sites.

| Station Name | Wanxian | Changshou | Fengjie | Qingxichang |
|:---|:---:|:---:|:---:|:---:|
| Maximum relative error | 1.49% | 3.54% | 1.73% | 2.41% |
| Minimum relative error | 0.00% | 0.00% | 0.00% | 0.00% |
| RMSE | 0.32% | 0.50% | 0.29% | 0.36% |
| Mean value of error | 0.22 | 0.45 | 0.089 | 0.17 |
| Variance of error with measured value | 0.63 | 1.87 | 0.63 | 0.85 |

**Table 2.** The Manning coefficient of the river.

| Section | Manning Coefficient |
| --- | --- |
| Zhutuo, Beibei~Cuntan | 0.030~0.038 |
| Cuntan~Changshou | 0.038~0.044 |
| Changshou, Wulong~Qingxichang | 0.042~0.046 |
| Qingxichang~Zhongxian | 0.035~0.038 |
| Zhongxian~Wanxian | 0.030~0.038 |
| Wanxian~Fengjie | 0.038~0.065 |
| Fengjie~The Three Gorges | 0.067~0.078 |

After completing parameter calibration for the hydrodynamic model, the data in 2017 was used to verify the accuracy of the model parameter calibration, and the results are shown in Figure 7.

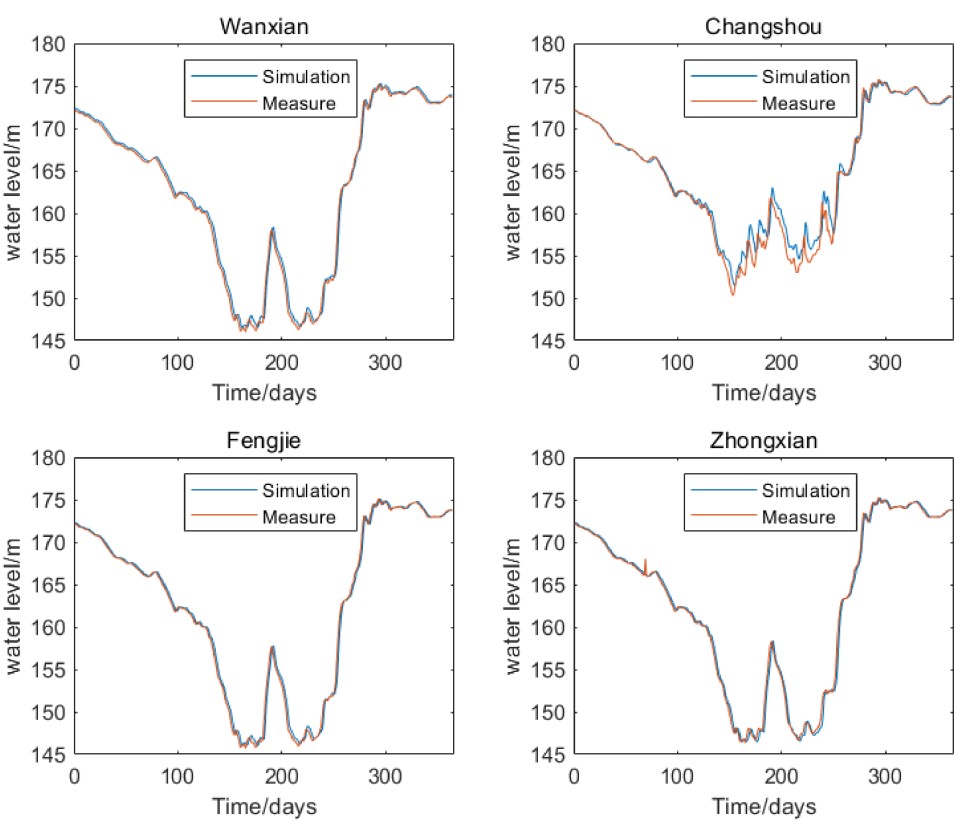

**Figure 7.** Comparison of the simulation and measured results during the verification period for each hydrological station.

It can be seen from Figure 7 and Table 3 that when considering interval inflow, the measured water level is consistent with the simulated water level, the average relative error at Changshou is only 0.44%, and the relative error for the whole reservoir area is less than 2%. As mentioned previously, Changshou station is located close to the tail of the reservoir. During the flood season, operation of the Three Gorges Reservoir is at the flood limit water level and upstream inflow is large, resulting in obvious dynamic storage capacity in the reservoir. The closer the dynamic storage capacity is to the tail of the reservoir, the greater its impact on reservoir operation activities, so the change is more intense. It can be seen from the figure that considering the interval inflow, the change in water level at Changshou station can be simulated more accurately, indicating that parameter calibration of the hydrodynamic model is more accurate.

**Table 3.** Comparison of simulation and measured results for different hydrological stations during the validation period.

| Station Name | Wanxian | Changshou | Fengjie | Zhongxian |
|---|---|---|---|---|
| Maximum relative error | 1.80% | 2.59% | 1.96% | 2.04% |
| Minimum relative error | 0.00% | 0.00% | 0.00% | 0.00% |
| RMSE | 0.30% | 0.44% | 0.26% | 0.26% |
| Mean value of error | 0.21 | 0.39 | 0.08 | −0.01 |
| Variance of error with measured value | 0.62 | 1.04 | 0.65 | 0.65 |

*4.3. The Model Establishment of Land Zone in Front of the Dam*

After parameter calibration, we established a hydrodynamic model between the upstream hydrological station and the end of the land zone in front of the dam. First, we needed to determine where the land zone ended in front of the dam, so we revisited the work of Lu [24].

Using measured data from 2014 to 2018, changes in the water surface profile for the Three Gorges Reservoir during the flood season are shown in Figures 8 and 9.

The Three Gorges Reservoir has a huge storage capacity. For every 1 cm change in water level, the water volume in the reservoir can change by millions of cubic meters. From Figure 8 we can see that when the variation range for the water level exceeds 1 cm, the straight section in front of the dam is at its shortest between Wushan and Fengjie hydrological stations and can reach the confluence of Wujiang River at its longest.

The water surface profile during the main flood season is shown in Figure 9. When 1 cm is taken as the judgment standard, the straight section in front of the dam is at its shortest between Badong and Zigui hydrological stations, and at its longest near Qingxichang hydrological station.

We can conclude that the water surface line between the Three Gorges Dam and Badong hydrological station is absolutely horizontal. Thus, for the calculation of water balance, the land zone in front of the dam is taken between the Three Gorges Dam and Badong hydrological station.

Based on the above rules, during short-term fine operation of the reservoir, it is necessary to formulate the daily reservoir operation plan on an hourly scale. The change law for the land zone in front of the dam shows that as the distance from the dam site increases, the changes become more intense. Therefore, an absolute straight section was selected to be combined with the hydrodynamic model. Badong hydrological station is within the confines of the absolute straight level; thus, we decided to use the section between the Three Gorges Dam and Badong hydrological station as the land zone in front of the dam.

Based on the land zone in front of the dam, a hydrodynamic model was established for the region between Zhutuo hydrological station and Badong hydrological station. Using the new water balance model to deal with flood routing through the reservoir, the results were compared with the traditional model.

It can be seen from Figures 10 and 11 that after adopting the new water balance calculation model proposed in this paper, the output process of the inversion is relatively consistent with the measured output, but the error for the lag time method is large. It can be seen from Table 4 that the average relative error for the lag method is 25.9%, but the average relative error of the new water balance model is only 8.5%.

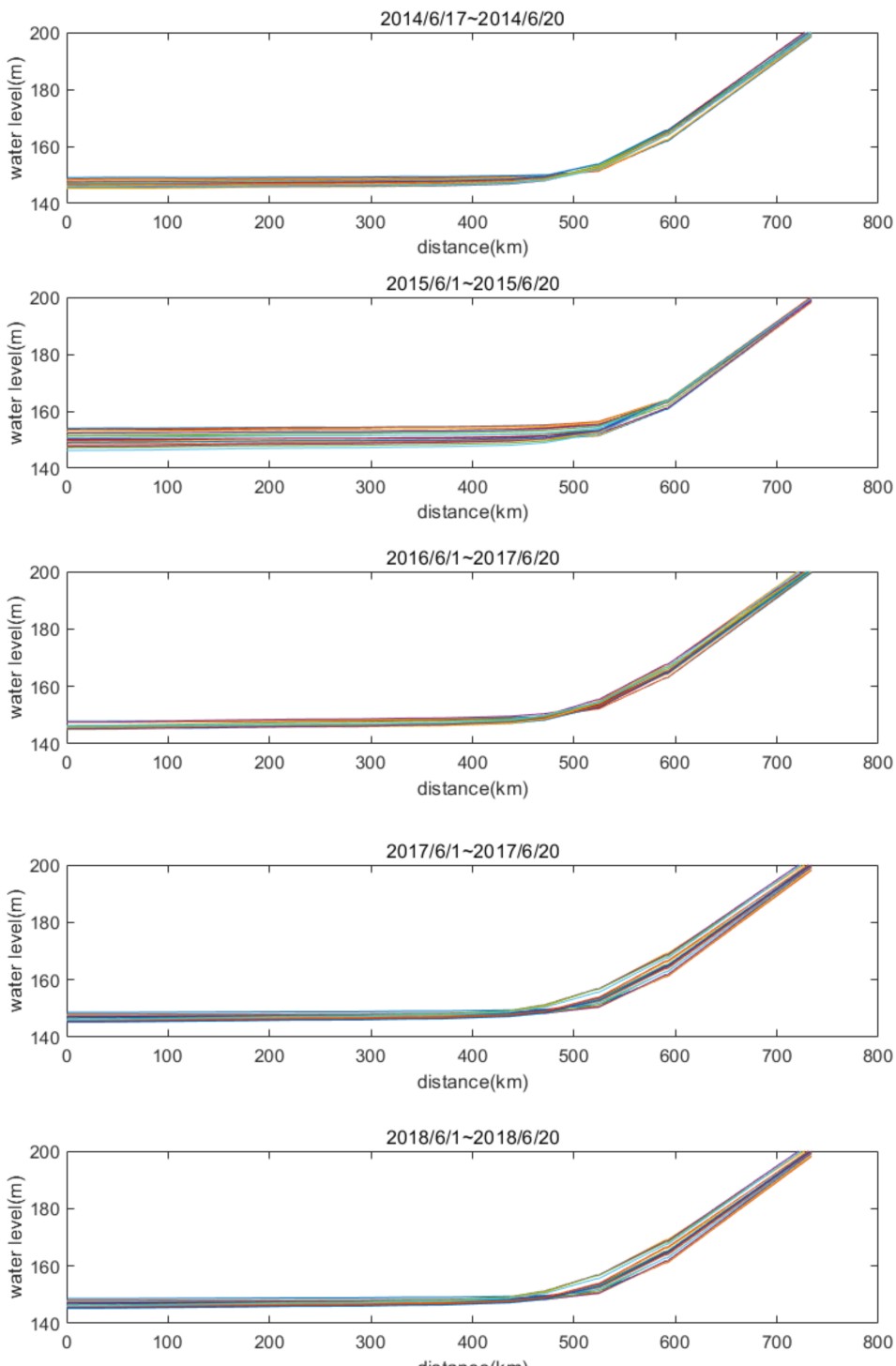

**Figure 8.** Water surface line during the early flood season from 2014 to 2018.

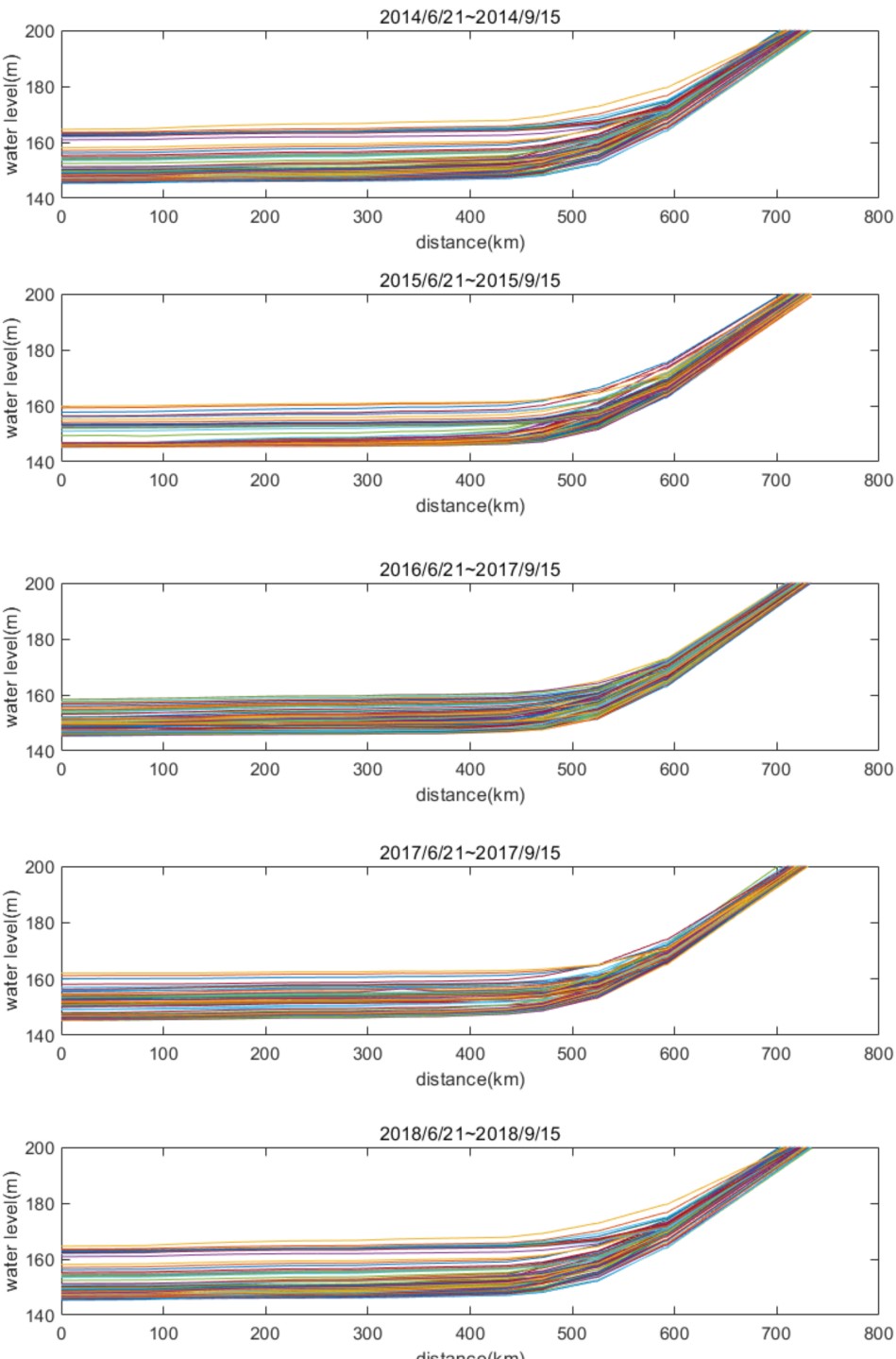

**Figure 9.** Water surface line during the main flood season from 2014 to 2018.

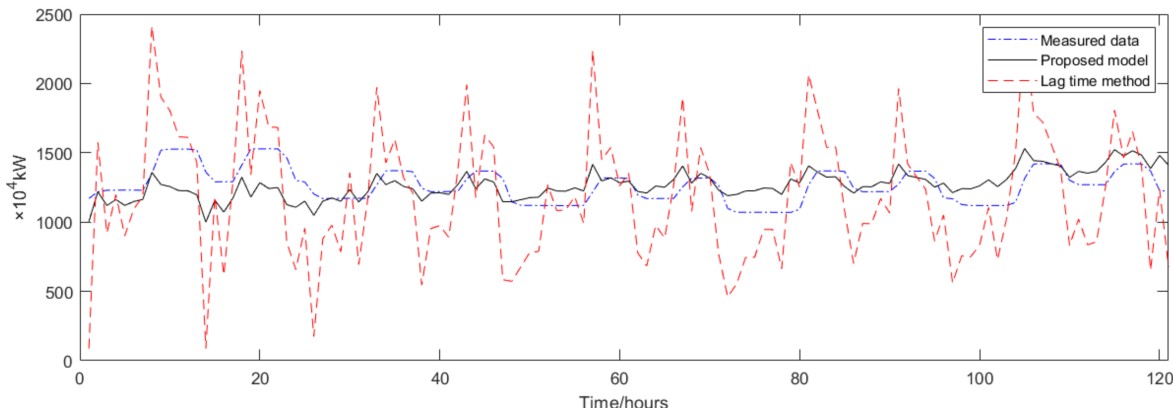

**Figure 10.** Comparison of the results from the traditional method with the proposed model.

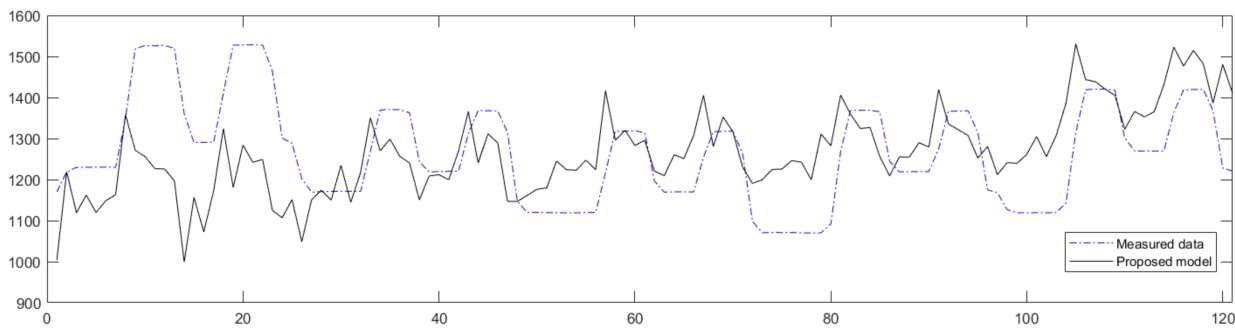

**Figure 11.** Comparison of the results from the proposed method with the measured data.

**Table 4.** Comparison of output inversion error between the time lag method and the new method proposed in this paper.

| Indicator | Lag Time Method | Proposed Model |
|:---:|:---:|:---:|
| RMSE | 25.90% | 8.50% |

The reason that the lag time method has a larger error than the new model is that the inflow of the reservoir cannot be calculated accurately, and the flatness of water flow and interval inflow are not considered. Moreover, the control body established by the time lag method cannot accurately calculate the water volume of the reservoir, which has the problem of water imbalance. The new method proposed in this paper combines a hydrodynamic model with the interval inflow, fully considering displacement and flatness in the process of water flow transmission, and the straight section in front of the dam can be used to accurately calculate the reservoir capacity, so the final result is more accurate.

The Three Gorges Dam has a large dynamic capacity during the flood season. Due to the influence of the dynamic storage capacity in the reservoir, calculation of the reservoir storage capacity by the time lag method will be prone to error, and there will be large random factors and great fluctuations that will affect short-term operation of the reservoir. When the model proposed in this paper is used to calculate the output, the output trend is close to the measured trend. However, the traditional time lag method is still used to calculate the interval inflow, and there will be a deviation between the output and the measured output due to random factors during short-term operation of the reservoir.

## 5. Conclusion and Future Work

This paper proposes a new water balance model that combines a hydrodynamic model with the land zone in front of the dam. The innovations of the proposed model are: (a) compared with the Muskingum method, the hydrodynamic model can provide more

accurate inflow for the water balance calculation; (b) by using the land zone in front of the dam, the storage capacity of reservoir can be calculated accurately; (c) compared with the lag time method, the results shows that the new water balance calculation model can calculate water balance more accurately. Using hourly scale data from 5 July 2019 to 10 July 2019, the new water balance calculation model was verified. The main conclusions can be summarized as follows:

(1) The hydrodynamic model was used to accurately calculate the inflow of the reservoir. Compared with the lag time method, it overcomes the disadvantages caused by only considering flow transposition and not considering flow attenuation. At the same time, it overcomes the disadvantage of the Muskingum method, which cannot consider the influence of reservoir jacking. A case study of the Three Gorges Dam showed that the average relative error for the calculation of inflow was only 2.79%, which can provide more accurate inflow data for refinement of the reservoir.

(2) Focusing on the problem of large errors in the calculation of reservoir storage capacity caused by dynamic capacity in the flood season, this paper introduces the land zone in front of the dam. The tail of the land zone in front of the dam and its upstream are all regarded as the river channel, and only the straight section in front of the dam is regarded as the reservoir so as to realize the accurate calculation of reservoir capacity.

(3) In this paper, the land zone in front of the dam was coupled with a hydrodynamic model for the first time, and a new water balance calculation model is proposed. By coupling a one-dimensional hydrodynamic model with the reservoir operation model in front of the dam, an output inversion calculation was carried out for the Three Gorges Dam as the research object. The research results show that the accuracy of the proposed method was significantly improved compared with the traditional reservoir operation model, since the RMSE of the new method was only 8.5%.

The model proposed in this paper can be used for the short-term refined operation of reservoirs to provide more accurate values for inflow and storage capacity. In the future, according to the variation law for the land zone in front of the dam, the dynamic land zone in front of the dam will be used to calculate reservoir capacity more accurately.

**Author Contributions:** S.Z.: Writing—original draft, Methodology. S.O., X.N.: data support. J.Z.: Funding acquisition. H.Q. (Hui Qin), J.H.: Supervision. H.Q. (Hongya Qiu): review. All authors have read and agreed to the published version of the manuscript.

**Funding:** This study is supported by the National Key Research and Development Program of China (2021YFC3200303) and the National Natural Science Foundation of China (No. U1865202, 51979113).

**Data Availability Statement:** The data of the hydropower station used in this study is confidential. Please contact the corresponding author if necessary. The water surface line in the research area can be found in the website of the Bureau of Hydrology, Changjiang Water Resource Commission. The water level data are updater every day.

**Acknowledgments:** This study is supported by National Key Research and Development Program of China (2021YFC3200303) and the National Natural Science Foundation of China (No. U1865202, 51979113). Special thanks are given to the anonymous reviewers and editors for their constructive comments.

**Conflicts of Interest:** The authors declare that they have no known competing financial interests or personal relationship that could have appeared to influence the work reported in this paper.

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
