# Peer review of "Water Balance Calculation Based on Hydrodynamics in Reservoir Operation"

_water, doi:10.3390/w14132001_

Round 1
Reviewer 1 Report
The authors improved their manuscript regarding the grammar. All my other comments were not taken into the account. I still insist on them.Author Response
I'm sorry that I can't modify the paper according to your requirements. Please tell me where to modify it. I will modify it according to your requirements

Reviewer 2 Report
Article 1735314 is devoted to the efficient operation of reservoirs. This problem is important in reservoir management. When operating a reservoir, calculating the water balance is an extremely important step. Currently, one of the main problems is that the inflow to the reservoir cannot be accurately calculated due to the rise of the reservoir, and the other is that the storage capacity of the reservoir cannot be accurately calculated due to the effect of dynamic capacity.
The novelty of the study is the authors propose to calculate the storage capacity of the reservoir, the land area in front of the dam is used, which is used as the boundary of the hydrodynamic model, and the hydrodynamic model is used to calculate the inflow of the reservoir in order to improve the accuracy of calculating the inflow to the reservoir.
The authors obtained the following practical results compared with the measured data, the standard deviation of the proposed model is 8.5%, while the standard deviation of the traditional model is 25.9%. The results of the application show that this study can provide technical support for the fine operation of reservoirs..
The authors were advised to slightly expand the introductory part. Supported by a couple of links.
Martyushev D. A. Improving the geological and hydrodynamic model a carbonate oil object by taking into account the permeability anisotropy parameter. Journal of Mining Institute. 2020. V.243. pp.313-318. https://doi.org/10.31897/PMI.2020.3.313
Potashev K.A., Akhunov R.R., Mazo A.B. (2022). Calculation of the flow rate between wells in the flow model of an oil reservoir using streamlines. Georesursy, 24(1), pp. 27–35. https://doi.org/10.18599/grs.2022.1.3
Martyushev D.A. Rock stress state influence on permeability of carbonate reservoirs. Bulletin of the Tomsk Polytechnic University, Geo Assets Engineering. 2020. V.331(8). pp.24-33. https://doi.org/10.18799/24131830/2020/8/2765
Yermekov R. I., Merkulov V. P., Chernova O. S., & Korovin M. O. (2020). Features of permeability anisotropy accounting in the hydrodynamic model. Journal of Mining Institute, 243, 299. https://doi.org/10.31897/pmi.2020.0.299

Author Response
The introduction of the thesis has been expanded according to the references you provided

Reviewer 3 Report
Line 117: is n or nm?
Line 137: operator-splitting technique. Pease give a reference.
Line 139: the semi-implicit method. Please give a reference.
Line 139: …to discretize the gradient of the water-level gradient. Please rephrase.
Line 143: … θ implicit coefficient which is between 0 and 1. Please give an explanation for the values [0,1].
Line: 159- 160: please split the equation in two lines because is difficult to read all the terms of the equation.
Line 160: something go wrong here. This line is not at the same horizontal.
Line 164: the Muskingum method’s control equation. Please give a reference.
Line 171: Eq (11)? Is that correct? The eq (11) is at line 179. Please check the numbering of equations.
Line 181: …ε is the time weighting factor which is in 0 and 1. Please give an explanation for the values [0,1]. Also in line 143 you use the same symbol θ. Is the implicit coefficient the same with the space weighting factor. Maybe it is better to change one of this two symbols.
Line 185: Assuming that θ is equal to 0.5: Please give an explanation on that and some references.
Figure 6: Please give an explanation for the small differences between simulation and measure for the Changshou case.
Table 1: More statistics are needed (stdev, mean, median ect)
Table 3: More statistics are needed (stdev, mean, median ect)
Figure 8: A legend is missing for the explanation of colors in Figure 8. Also for the 2015/6/1-2015/6/20 figure, please remove the icons of: distance, copy, paste, zoom in, zoom out.
The same for Figure 9.
Figure 10: Please give an explanation why the Lag time method gives higher values than the measured data. The proposed method gives similar results with the measured data. But the reviewer proposes to make an extra figure with the comparison of the measured data and the proposed data, because they are sizes of the same order of magnitude.
Have you compare the results of the proposed method with the results of the commercial code MIKE DHI? The reviewer believes that this comparison will greatly enhance the quality of your work.
For the reference 16: This reference is correct. DHI M? DenmarN? Please check
Author Response
Responses to Reviewers’ Comments and Suggestions
By Zhu et al.
Submitted to Water
Manuscript ID: water-1735314
_____________________________________________________________________
The authors are grateful to reviewers for their valuable comments and suggestions on the manuscript entitled “Study on fine reservoir operation based on hydrodynamic method”. The whole manuscript has been carefully revised and improved in accordance with the reviewers’ comments and suggestions. The rewritten sentences are highlighted in red in the revised manuscript; the added texts are underlined; and the deleted texts are indicated by strikethrough formatting. The main revisions are summarized as follows:
Point 1: Line 117: is n or nm?
Response 1: nm is correct.
Point 2: Line 137: operator-splitting technique. Pease give a reference
Response 2: The reference has been given which is “Hu, D., Lu, C., Yao, S., Yuan, S., Zhu, Y., Duan, C., & Liu, Y. (2019). A prediction–correction solver for real-time simulation of free-surface flows in river networks. Water, 11(12), 2525”
Point 3: Line 139: …to discretize the gradient of the water-level gradient. Please rephrase.
Response 3:The sentence has been changed to “To eliminate the stability restriction caused by the gravity waves, the semi-implicit method is introduced to discretize the water-level gradient term”
Point 4: Line 143: … θ implicit coefficient which is between 0 and 1. Please give an explanation for the values [0,1].
Response 4:θis an empirical parameter used to determine the display discrete part and the implicit discrete part
Point 5: Line: 159- 160: please split the equation in two lines because is difficult to read all the terms of the equation.
The lines have been split in two lines.
Response 5:After discretizing the continuity and momentum equation, we need to solve velocity-pressure coupling problem. For one river (see Figure 1) velocity-pressure is performed by substituting Equation (4) to Equation (5) then we can get equation (6).
(6)
where
and
.
Point 6: Line 160: something go wrong here. This line is not at the same horizontal.
Response 6: This line has been made at the same horizontal
Point 7: Line 164: the Muskingum method’s control equation. Please give a reference.
Response 7: The reference is “Todini, E. (2007). A mass conservative and water storage consistent variable parameter Muskingum-Cunge approach. Hydrology and Earth System Sciences, 11(5), 1645-1659.”
Point 8: Line 171: Eq (11)? Is that correct? The eq (11) is at line 179. Please check the numbering of equations.
Response 8: The equation is wrong, the correct equation is as follow:
(11)
Point 9: Line 181: …ε is the time weighting factor which is in 0 and 1. Please give an explanation for the values [0,1]. Also in line 143 you use the same symbol θ. Is the implicit coefficient the same with the space weighting factor. Maybe it is better to change one of this two symbols.
Response 9: I have change the the symbols in the equation 11
(11)
(12)
where the μ is the space weighting factor which is between 0 and 1 and ε is the time weighting factor which is between 0 and 1. Equation (11) and (12) can lead to the following first order approximation of the kinematic wave Equation.
Point 10: Line 185: Assuming that θ is equal to 0.5: Please give an explanation on that and some references.
Response 10: The reference is “Todini, E. (2007). A mass conservative and water storage consistent variable parameter Muskingum-Cunge approach. Hydrology and Earth System Sciences, 11(5), 1645-1659.”
Point 11: Figure 6: Please give an explanation for the small differences between simulation and measure for the Changshou case.
Response 11: I have add explanation for the Changshou case, the explanation is as follow:
It can be seen from the figure that the error is more obvious in the flood season, and the simulated value and the measured value are in good agreement in the non-flood season. This is because the manning coefficient is different due to the different water levels of the Changshou hydrological station in flood season and non flood season, which affects the simulation results.
Point 12: Table 1: More statistics are needed (stdev, mean, median ect); Table 3: More statistics are needed (stdev, mean, median ect)
Response 12: Table 1 and Table 3 have added more statistics
Table 1 Statistical comparison results between simulated and measured data at different hydrological sites
Station name |
Wanxian |
Changshou |
Fengjie |
Qingxichang |
Maximum relative error |
1.49% |
3.54% |
1.73% |
2.41% |
Minimum relative error |
0.00% |
0.00% |
0.00% |
0.00% |
RMSE |
0.32% |
0.50% |
0.29% |
0.36% |
Mean value of error |
0.22 |
0.45 |
0.089 |
0.17 |
Variance of error with measured value |
0.63 |
1.87 |
0.63 |
0.85 |
Table 3 Comparison of simulation and measured results of different hydrological stations in the validation period
Station name |
Wanxian |
Changshou |
Fengjie |
Zhongxian |
Maximum relative error |
1.80% |
2.59% |
1.96% |
2.04% |
Minimum relative error |
0.00% |
0.00% |
0.00% |
0.00% |
RMSE |
0.30% |
0.44% |
0.26% |
0.26% |
Mean value of error |
0.21 |
0.39 |
0.08 |
-0.01 |
Variance of error with measured value |
0.62 |
1.04 |
0.65 |
0.65 |
Point 13: Figure 8: A legend is missing for the explanation of colors in Figure 8. Also for the 2015/6/1-2015/6/20 figure, please remove the icons of: distance, copy, paste, zoom in, zoom out.
Response 13: The icons has been removed. The legend has been added in Figure 8 and 9.
Figure 8 Variation of water surface level in early flood season from 2014 to 2018
Figure 9 Variation of water surface level in main flood season from 2014 to 2018
Point 13: Figure 10: Please give an explanation why the Lag time method gives higher values than the measured data. The proposed method gives similar results with the measured data. But the reviewer proposes to make an extra figure with the comparison of the measured data and the proposed data, because they are sizes of the same order of magnitude.
Response 13: The explanation have been added as follow:
The Three Gorges reservoir has large dynamic capacity in flood season. Due to the influence of the dynamic storage capacity of the reservoir, the calculation of the reservoir storage capacity by the time lag method will bring great storage capacity error, and there will be large random factors in the short-term operation of the reservoir, so the calculation by the time lag method will bring great fluctuations. When the model proposed in this paper is used to calculate the output, the output trend is close to the measured trend. However, the traditional time lag method is still used to calculate the interval inflow, and there is a deviation between the output and the measured output due to the random factors in the short-term operation of the reservoir.
The extra figure is as follow:
Figure11 Comparison results of the proposed method and the measured data

Round 2
Reviewer 3 Report
The paper is accepted in the present form.
Author Response
Responses to Reviewers’ Comments and Suggestions
By Zhu et al.
Submitted to Water
Manuscript ID: water-1735314
_____________________________________________________________________
The authors are grateful to reviewers for their valuable comments and suggestions on the manuscript entitled “Study on fine reservoir operation based on hydrodynamic method”. The whole manuscript has been carefully revised and improved in accordance with the reviewers’ comments and suggestions. The rewritten sentences are highlighted in red in the revised manuscript; the added texts are underlined; and the deleted texts are indicated by strikethrough formatting. The main revisions are summarized as follows:
Point 1: The paper is accepted in the present form.
Response 1: Thank you for accepting my thesis

This manuscript is a resubmission of an earlier submission. The following is a list of the peer review reports and author responses from that submission.
Round 1
Reviewer 1 Report
Paper presents a very quality hydrodynamic model for the sizing of the open accumulations, with regards to the changing of the inflow and a dynamic water level (flow) of the accumulation. Literature review is provided at the biggest level, with actual state-of-the-art of the sizing methods. Obtained results, i.e. comparison by the actual and simulated and observed data shows very good overlapping. I am proposing a minor revision. Here are reasons for my decision. -authors should apply some time series analysis methods. For example, IPTA in order to test homogeneity of the analyzed and observed time series. Also, natural or anthropogenic irregularities should be determined if they exist by application of the RAPS method. -line 399-400: authors should define what they mean by ''accuracy''. -Authors should comment on the impact of the precipitation, infiltration and evaporation on the water amount in the accumulation. -many of the figures are blurry.Author Response
The English language and style have been checked by the mdpi author services. For proof of modification of the thesis, see attachment.
The Input hydrology data for hydrodynamic model is measured data. The trend of input data is shown in fiugre which is in the attachment. The terrain data is measured data. Although there are abrupt changes in the data, the hydrodynamic model used in this paper can better deal with these problems.
The impact of the precipitation, infiltration and evaporation on the water amount in the accumulation have been considered in the hydrodynamic by using the source term in the hydrodynamic.

Reviewer 2 Report
Title of paper “Study on fine reservoir operation based on hydrodynamic method” does not reflect the purpose of the paper since the authors state in L16-18: “The contributions and novelty of this paper are: (a) the proposed model CAN COMBINE the hydrodynamic model with the water balance calculation model, which makes the calculation of the inflow more accurately.
Language is a major concern. There are many grammar inconsistencies. Not only language. Just trying to follow the discourse is not easy. For instance the Abstract becomes repetitie and intricated at the very beginning with: “Reservoir operation plays an important role in reservoir management. In reservoir operation, water balance calculation is a very important step. At present, one of the main challenges is the reservoir inflow can not be calculated accurately due to the jacking of the reservoir, the other is the reservoir capacity can not be calculated accurately because of the influence of dynamic storage capacity. In order to overcome the above problems, the land zone in front of the dam is used to calculate the reservoir capacity, which is used as the boundary of the hydrodynamic model, and the hydrodynamic model is used to calculate the reservoir inflow, so as to improve the calculation accuracy of reservoir inflow….”
The Abstract ends with the statement “The application results indicate that this study can provide technical support for fine operation of reservoirs.” which is unclear according to the whole paper or the very final part of the Conclusion “The model proposed in this paper can be used for the calculation of the short-term refined operation of reservoirs to provide more accurate calculation of inflow and storage capacity. In the future, according to the variation law of the land zone in front of the dam, the dynamic land zone in front of the dam will be used to calculate the reservoir capacity more accurately.”
Paper has serious drawbacks:
- All equations should be explained and avoid missprints given of coefficients and their meaning. What is n_m in Eq (2). Is g given in m^2/s in L116? What are the control volumes mentioned in L126?
- Sections 3 and 4 contain a lot of data without explaining the sources.
- It is weird to consider data from 2005,2016,2017 in the very well studied area of the Three Gorges Reservoir to get parameters and validate them according to an hourly scale data from July 5, 2019 to July 10, 2019 to verify the new water balance calculation model.
- DHI MIKE is recalled but not clearly explained how it is implemented.
There is no clear explanation of the link between four above items within the paper.
Paper has several sentences that are hard to understand because of grammar errors and/or missing connective words. Some specific misprints:
L74 Mike?
L82 high robust high efficient
L89 reservoir’s capacity
L91 big error in calculate
L94 the other is
L104 the overview the models
L119 the a
Figure 1 is impossible to understand
Figure 3 is hard to understand… what does t<T mean?
The improvement is justified just by comparing graphs obtained from data with the ones obtained by the model proposed. There is no possibility to replicate their experience.
L457 Data reference must be completed
This is not an exhaustive list of missprints. Paper requires a full revision by a native speaker.
Author Response
Point 1: Language is a major concern. There are many grammar inconsistencies. Not only language. Just trying to follow the discourse is not easy.
Response 1: The grammar have been checked by the mdpi author service.
Point 1: All equations should be explained and avoid missprints given of coefficients and their meaning. What is n_m in Eq (2). Is g given in m^2/s in L116? What are the control volumes mentioned in L126?
Response 1: These question have been modify in the revision. The control volume is the area between the two dotted line. The control volume which is used to discretize the control equation of the hydrodynamic see figure 1 in the attachment.
Point 2: Sections 3 and 4 contain a lot of data without explaining the sources.
Response 2: The data used in section 3 and section 4 are measured data including the hydrological data and topographic data.
Point 3: DHI MIKE is recalled but not clearly explained how it is implemented.
Response 3: DHI Mike is not used in this paper. It is only reviewed in the introduction. In the introduction, it is also reviewed that DHI Mike uses the finite difference method to discretize the governing equations
Point 4: It is weird to consider data from 2005,2016,2017 in the very well studied area of the Three Gorges Reservoir to get parameters and validate them according to an hourly scale data from July 5, 2019 to July 10, 2019 to verify the new water balance calculation model.
Response 4: The 2005 data were used to compare the difference between the Muskingum method and the hydrodynamic method. 2016 is a wet year. Using the data in 2016 can well reflect the roughness of the study area. In the calculation of hourly scale, the change of water level and flow will not be very large, so the data calibrated by daily scale can be used for calculation

Round 2
Reviewer 2 Report
The authors have used MDPI Authors service and paper has improved.
According to L84, results are based on Lu [28] and that refernece is not cited in standard way with all data. It is not possible to consult it.
It is the basis to combine the hydrodynamic model with the water balance calculation and the land zone in front of the dam to make the calculation of the reservoir capacity. Reference [28] need nore datal.
All calculations in Section 4.3 come from rough assumptions which in turn are based on [28] as the authors honestly recall in L397, L406.
For coherence with the rest -> Address 3 -> Changjiang Institute of Survey, Planning, Design and Research Corporation,